# Capturing variation impact on molecular interactions in the IMEx Consortium mutations data set

The IMEx Consortium Curators[#], N. del-Toro[1], M. Duesbury[1], M. Koch[1,2], L. Perfetto[1], A. Shrivastava [1], D. Ochoa [1], O. Wagih[1,3], J. Piñero [4], M. Kotlyar[5], C. Pastrello [5], P. Beltrao [1], L.I. Furlong[4], I. Jurisica [5,6], H. Hermjakob [1,7], S. Orchard[1] & P. Porras [1]

The current wealth of genomic variation data identified at nucleotide level presents the challenge of understanding by which mechanisms amino acid variation affects cellular processes. These effects may manifest as distinct phenotypic differences between individuals or result in the development of disease. Physical interactions between molecules are the linking steps underlying most, if not all, cellular processes. Understanding the effects that sequence variation has on a molecule's interactions is a key step towards connecting mechanistic characterization of nonsynonymous variation to phenotype. We present an open access resource created over 14 years by IMEx database curators, featuring 28,000 annotations describing the effect of small sequence changes on physical protein interactions. We describe how this resource was built, the formats in which the data is provided and offer a descriptive analysis of the data set. The data set is publicly available through the IntAct website and is enhanced with every monthly release.

[1] European Bioinformatics Institute (EMBL-EBI), European Molecular Biology Laboratory, Wellcome Genome Campus, Hinxton CB10 1SD, UK. [2] Novartis Institutes for BioMedical Research (NIBR), Maulbeerstrasse 66, 4058 Basel, Canton of Basel-Stadt, Switzerland. [3] Deep Genomics, MaRS Centre, 661 University Ave, Suite 480, Toronto, ON M5G 1M1, Canada. [4] Research Programme on Biomedical Informatics (GRIB), Department of Experimental and Health Sciences (DCEXS), Hospital del Mar Medical Research Institute (IMIM), Universitat Pompeu Fabra (UPF), 08003 Barcelona, Spain. [5] Krembil Research Institute, Data Science Discovery Centre for Chronic Diseases, University Health Network, 5KD-407, 60 Leonard Avenue, Toronto, ON M5T 0S8, Canada. [6] Departments of Medical Biophysics and Computer Science, University of Toronto, Toronto M4B 1B5, Canada. [7] State Key Laboratory of Proteomics, Beijing Proteome Research Center, National Center for Protein Sciences, Beijing Institute of Life Omics, 102206 Beijing, China. [#]A full list of consortium members appears at the end of the paper.  Correspondence and requests for materials should be addressed to P.P. (email: pporras@ebi.ac.uk)

Cells process information and respond to their environments through dynamic networks of molecular interactions, where the nodes are bio-molecules (e.g. proteins, genes, metabolites, miRNAs) and edges represent functional relationships, including physical protein–protein interactions, transcriptional regulation, genetic interactions and gene/protein modifications. Comprehensive and systematic characterization of these networks is essential to gain a full understanding of complex biological processes, of how cells behave in response to specific cues, and of how individual components of the network contribute to phenotype, in physiological, pathological or synthetic conditions.

Interactions between molecules may be inherently stable and essentially irreversible, resulting in the formation of stable macromolecular complexes, or weak transient interactions characterized by a dissociation constant (KD) in the micromolar range and a lifetime of seconds. A change to a single amino acid in a protein chain can be enough to disrupt a protein binding site and may then alter the composition of a sub-network of transient binders or the formation of a protein complex. A variant leading to the inactivation of a protein kinase molecule may result in widespread disruption of post-translational phosphorylation events and the rewiring of related signalling networks. Many diseases are caused by specific mutations, and prognosis or response to treatment is frequently mutation-specific. The study of how mutations affect molecular interactions is thus of extreme interest since it can help ascertain the role of specific protein residues on the universal function of molecular binding. Several studies[1–4] have explored the impact of disease-related variation in molecular interaction networks, using structural studies and computational predictions to attempt to both identify variation-affected interfaces and predict the effect of specific variants on interactions. These studies suggest that interaction interfaces contain a significantly higher rate of disease-related variants than the rest of the molecule and that variant location in these interfaces can determine disease specificity.

Despite available high-throughput interaction screening platforms, the experimental validation of these variation effect predictions on a systems-scale remains a major challenge. However, these data can be found reported in the literature but difficult to search and concatenate. Researchers have for many years been examining the effect of single, or multiple, induced point mutations on both binary and n-ary interactions in small-scale experiments. Targeted changes to the amino acid sequence of a protein have been engineered, largely by site-directed mutagenesis, with the aim of mimicking known variants[5,6], removing known, or predicted, post-translational modifications[7,8], disrupting regions required for protein stability or altering the properties of protein binding domains[9,10], and their effects of the interaction of interest monitored. It has been the work of the IMEx Consortium[11] to capture such information into a single data set and thus make it available for researchers to re-use and re-analyse. IMEx Consortium annotators follow a detailed curation model, capturing not only full details of the experiment (including interaction detection method, participant identification method and the host organism) but also a description of the constructs used. This may include the coordinates of deletion mutants used to derive a minimum binding domain and also the effect of point mutations. Databases in the Consortium perform detailed, archival curation of published literature and also receive pre-publication data through direct submissions. This close collaboration with data producers often entails access to unpublished details in the data, such as experiments reporting mutations that have no effect on interactions, which enables the capture of added value for the scientific community.

Here we describe the largest literature-derived data set, to our knowledge, capturing the effect of sequence changes on interaction outcome. We discuss how the data set was generated and how it is maintained by the EMBL-EBI IntAct team. We also provide an initial analysis of the data set, highlighting its overlap with genomic variation data, discussing possible biases and exploring its potential as a benchmarking tool for variant effect prediction tools.

## Results

**Data curation and quality control**. The IMEx Consortium databases have been collecting point mutation data for over 14 years, which has resulted in a sizeable data set of almost 28,000 fully annotated events (www.ebi.ac.uk/intact/resources/datasets#mutationDs). The IMEx resources curate interaction data into structured database fields, and from there into community standard interchange formats, and each observation is described using controlled vocabulary terms. Mutations are mapped to the underlying protein sequence in UniProtKB and updated in line with changes to that sequence, to ensure that they stay mapped to the correct amino acid residue with every proteome release.

In order to make the mutations data set more accessible to the biomedical scientist, the Consortium has released it in a tab-delimited format (Table 1 and Supplementary Table 1), which includes details of the position and the amino acid change of the mutation, the molecules in the interaction and the effect of the mutation on the interaction, as well as additional fields containing contextual information.

Additionally, a data-update pipeline has been specifically developed to ensure the accuracy of the annotation of mutation events as interaction participant features (Supplementary Figure 1). The construction of this pipeline has been made possible by the creation of specific fields capturing sequence changes in our recently developed standard format PSI-MI XML3.0[12]. It is run in coordination with the IntAct database monthly protein update procedure, which ensures synchronization with UniProtKB[13] and automatically shifts feature positions if there are changes in referenced protein sequences. The pipeline is applied to the entire data in the IntAct database (www.ebi.ac.uk/intact), in which all IMEx data, and also legacy data generated by the IntAct, MINT, DIP and UniProt curation groups is housed (see Supplementary Information, sections 'Initial re-curation of mutation data in IMEx' for details on re-annotation and 'Automated quality control pipeline for mutation entries in IMEx' for data-update procedures). The mutation data-update pipeline will continue to be run in quality control mode with every release of IntAct to ensure the mutations data set is kept entirely up-to-date with UniProtKB.

**Data set statistics**. The full IMEx mutations data set contains 27,868 fully annotated events in which a sequence change has been experimentally tested in an interaction experiment. All this information has been manually curated, representing over 33,000 person-hours' worth of biocurators' work, and it is continuously growing with on-going IMEx curation activities. The 4353 proteins annotated come from 297 different species, with over 60% of the events annotated in human proteins and roughly 90% annotated in seven main model organisms (see Table 2).

In total, 13,926 interaction evidences are annotated with differentially reported effects, using the PSI-MI controlled vocabulary. Most of the effects reported are of a 'deleterious' nature, either disrupting (10,976 annotations, 39.3%) or decreasing the interaction (8553 annotations, 30.7%), but there is a significant number of interactions that are either strengthened (2256 annotations, 8.1%) or caused (188 annotations, 0.7%) by the mutation when compared with the wild-type sequence (Fig. 1a). The data set also includes those mutations that were

**Table 1 Overview of the IMEx mutations data set downloadable flat file**

| Feature AC | Feature short label | Feature range(s) | Original sequence | Resulting sequence | Feature type | Feature annotations | Affected protein AC & details[a] | Interaction participants & details[a] |
|---|---|---|---|---|---|---|---|---|
| EBI-10828532 | p. Arg725Glu | 725–725 | R | E | mutation (MI:0118) | MI:0612 (comment): Disrupts association with VPS33A and decreases association of VPS18 | See Supplementary Table 1 | See Supplementary Table 1 |
| EBI-985220 | p.Ile114Gly | 114–114 | I | G | mutation increasing (MI:0382) | | " | " |
| EBI-4370347 | p. [Asn31His; Ala60Val] | 31–31 | D | H | mutation increasing (MI:0382) | - (kd): 11e-9M | " | " |
| EBI-4370347 | p. [Asn31His; Ala60Val] | 60–60 | A | V | mutation increasing (MI:0382) | - (kd): 11e-9M | " | " |
| EBI-10688294 | p.Thr2Ala | 2–2 | T | A | mutation decreasing rate(MI:1130) | | " | " |
| EBI-9635600 | p. Cys_Ser215-216Ala_Ala | 215–216 | CS | AA | mutation disrupting (MI:0573) | | " | " |

Each mutation annotation is identified with a unique accession ('Feature AC') and is described with an HGVS-compliant short label ('Feature short label'), plus sequence coordinates and amino acid replacement details ('Feature range(s)', 'Original sequence', 'Resulting sequence'). The effect the mutation has on the interactions is listed using the PSI-MI controlled vocabulary ('Feature type'). Each line in the file represents a single continuous section of sequence affected, with mutations spread in multiple positions represented in several lines and sharing the same 'Feature AC'. Annotations for complex effects that cannot be captured via PSI-MI CV and kinetic parameters are also recorded when available ('Feature annotations'). [a]These are placeholders for additional columns containing information about the affected proteins and interactions. These are available in the fully expanded version of this table in Supplementary Table 1

**Table 2 Summary statistics per organism**

| Organism | Annotated events | Sequence changes | Affected proteins | Affected interactions | Source publications |
|---|---|---|---|---|---|
| *Homo sapiens* | 16,861 | 7955 | 2095 | 8268 | 2219 |
| *Mus musculus* | 2236 | 1406 | 482 | 1248 | 509 |
| *Saccharomyces cerevisiae* | 2029 | 1144 | 363 | 1069 | 326 |
| *Arabidopsis thaliana* | 1172 | 546 | 187 | 590 | 189 |
| *E. coli* (strain K12) | 979 | 614 | 143 | 374 | 148 |
| *Rattus norvegicus* | 562 | 354 | 142 | 341 | 160 |
| *Drosophila melanogaster* | 359 | 232 | 100 | 214 | 92 |
| Others (290 species) | 3670 | 2396 | 841 | 1951 | 855 |
| Totals | 27,868 | 14,647 | 4353 | 13,926 | 4182 |

experimentally tested but found to have no effect over the interaction (3057 annotations, 11%) and 'undefined' mutations that were present in constructs used in the experiment but where the comparison with the wild type reference is either absent or not possible (2838 annotations, 10.2%). It is important to note that the 'causing' and 'no effect' mutation effect categories have been only recently adopted into the controlled vocabulary and captured by the biocurators, so they have a much lower number of annotations and are not directly comparable with the other categories.

Protein–protein interaction (PPI) experiments reporting this type of data have been steadily increasing in the last 20 years, with over 4100 publications containing data pertaining to mutated protein sequences curated by the IMEx Consortium. However, the fraction of PPIs in which a mutated version of a protein has been reported remains relatively low (Fig. 1b). The majority of the interactions where a mutated protein was involved were detected using either affinity chromatography-related methods (such as co-immunoprecipitations or pull-downs) or by complementation assays based on transcriptional reporters, mainly variations of the yeast two-hybrid method (see Fig. 1c). Most of our data set comes from the curation of small-scale papers each reporting only a few mutations (Fig. 1d). Around 99% of the publications (4173

contain less than 100 mutation annotations and represent 80% of the annotations (22,218). Only 8 publications contain over 100 annotations, with one of them describing over 4000 events, a study in which the authors systematically tested large numbers of variants and their effect on interactions[5]. Recording large-scale data sets such as this one has been enabled by the development of the flexible PSI-MI XML3.0 format cited above.

Currently, the only resources that represent the impact of amino acid substitutions on binding events are the SKEMPI 2.0 database[14], UniProtKB and IMEx Consortium member databases through IntAct (see Table 3 for a detailed comparison). Of these resources, IMEx is the biggest and the only one that can provide easily accessible, systematically described, up-to-date annotations. UniProtKB mutagenesis annotations record whether a change in sequence affects an interaction, but the experimental context is not captured and the effects are described in a semi free-text field that is difficult to parse. SKEMPI offers a detailed overview of sequence change effects on binding derived from in vitro experiments, recording changes in affinity and other kinetic parameters for protein complexes with available structural data in the Protein Data Bank (PDB)[15]. Only very specific interaction detection methods, using purified proteins, are considered, which limits its scope.

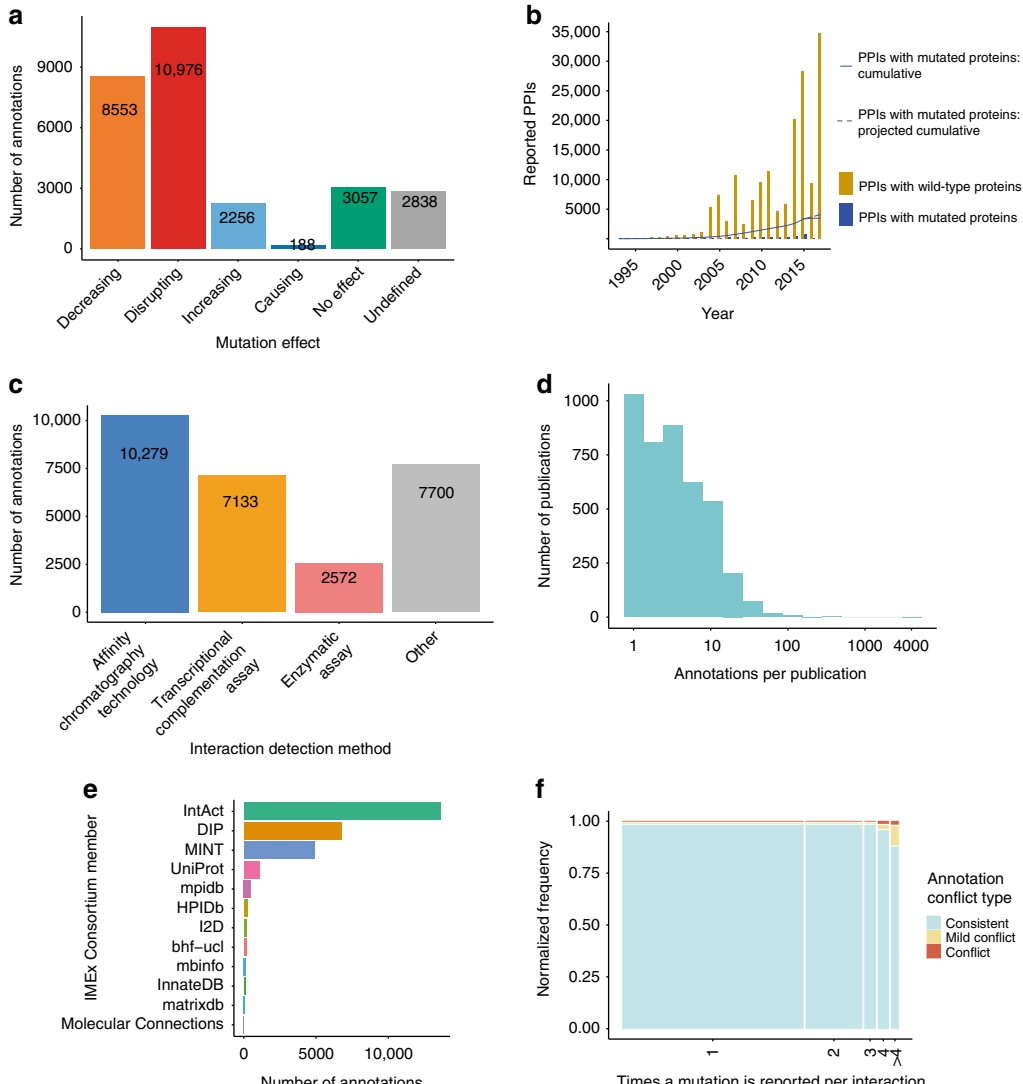

**Fig. 1** IMEx mutations data set overview. **a** Number of annotations by effect type; **b** Increase of reported protein interactions involving wild type and mutated proteins over time. Bars represent total number of PPIs reported each year (wild type proteins in gold, mutated proteins in blue). Lines represent the cumulative sum of PPIs with mutated proteins, with the solid line representing the actual cumulative trend and the dashed line showing a projection for the last 3 years; **c** Distribution of the number of mutation annotations by interaction detection method; **d** Distribution of the number of mutation annotations captured per publication. The number of annotations per publication is shown on a log scale; **e** Number of mutation annotations per database of origin; **f** Internal consistency of repeatedly reported mutations. 'Conflict' cases are those in which the effects reported are antagonistic (e.g. 'disrupting' vs 'increasing'). 'Mild conflict' cases are those in which the mutation is sometimes reported as having some effect vs others in which there is no detectable effect

There is limited overlap between these resources, with only 4 publications and 44 reported sequence changes found in all three. We manually assessed the consistency of the annotations for these 4 publications and found a number of inconsistencies caused by different curation practices. Most notably, SKEMPI 2.0 reports mutation coordinates using the chains as annotated in PDB, which often differs from the actual amino acid positions as reported in UniProt entries, used by IMEx and UniProt. We also found cases where mutations with little or no effect over interaction outcome were not reported in the IMEx data set, a result of early curation guidelines, which did not require the annotation of a mutation unless an effect was clearly shown. Additionally, UniProt and SKEMPI 2.0 report mutations for which the actual evidence was originally generated in a publication referenced in the one attached in the record. According to IMEx guidelines, all mutation annotations must be referred to the original publication. Finally, some annotations

were just missing in one or more of the resources for no obvious reason, although we could attribute some cases to non-parseable UniProt records. Full details of our manual assessment can be found in Supplementary Data 1.

The IMEx Consortium is currently formed by 11 groups, each one with their own area of interest, that have agreed to use the same curation standards and data representation download formats. All members of the consortium[16–23] use the curation platform provided by the IntAct team at EMBL-EBI. Figure 1e shows the number of events annotated by each data resource. Large databases such as IntAct, DIP and MINT, with an exclusive focus on interaction data curation, have produced the majority of the annotations, but a sizeable part of the data set has been entered by other, domain-specific, members of the Consortium.

According to the IMEx schema and curation policy, interaction evidence, rather than interacting pairs of molecules, is the focus of the data representation. This results in the curation of multiple

**Table 3 Resources reporting mutation effect over interaction**

|  | IMEx mutations data set | UniProtKB mutagenesis annotations | SKEMPI 2.0 |
|---|---|---|---|
| Annotations | 27,868 (16,861 human) | 8620 (5265 human) | 6108 (2743 human) |
| Sequence changes | 14,647 (7955 human) | 8297 (5043 human) | 3126 (1724 human) |
| Publications | 4182 | 3344 | 237 |
| Species | 297 | 93 | 55 |
| Description of variation effect | Structured controlled vocabulary | Free text, controlled syntax | Structured tabular representation |
| Referenced to original publication | Yes | Yes | Yes |
| Interaction experimental context | Fully captured | Not captured | Only kinetics |
| Kinetic parameters associated with variation | Yes, if available (very few cases) | No | Yes |
| Up-todate with proteome builds | Yes | Yes | No (referenced to PDB) |
| Active curation | Yes | Yes | Uncertain (last update 2018) |
| Accessibility | Table accessible through ftp, website, standard formats for interaction representation | UniProt ftp, website, Proteins API | Website, downloadable CSV table |
| Website | www.ebi.ac.uk/intact/resources/datasets#mutationDs | www.uniprot.org | https://life.bsc.es/pid/skempi2/ |

distinct pieces of evidence describing the same interacting pairs and offers a way to weight how well characterized is a given interacting group of molecules. It also enables us to capture separate experiments where different sequence variants are tested for their effect on an interaction. Most of the proteins in the data set have a low number of associated mutations, with most proteins having less than 15 annotations (Supplementary Figure 2a) and 5 or less sequence changes (Supplementary Figure 2b). There is a greater depth of information available for human proteins, since the relative amount of human data vs other species increases with the number of annotations per protein.

The IMEx evidence-centric curation model also makes it possible to check whether the same mutation has been tested on identical interacting molecules using different interaction detection methodologies (or by different research groups) and whether the outcome of the mutations has been consistent in all these experiments. In Fig. 1f we show that the majority of the mutations have only been annotated once (tested in one experiment only). In those cases where there have been multiple instances of evidence testing, the results appear to be highly consistent, with only a small number of cases identified for which conflicting results have been reported. For 7212 cases where the effect of a mutation on an interface was tested 2 or more times, only 90 (1.3%) show different effects, and only 19 cases (0.3%) reported antagonistic effects. We carefully checked antagonistic cases and found that in 17 out of 19 cases, the reason for these apparently contradictory results were mutant forms tested in experimental setups that provide fundamentally different types of information. The most common case is when the mutant and wild-type versions of a protein were tested for enzymatic activity and for binding in separate assays. For example, *Bacillus subtilis* *SufU* C41A variant forms an hetero-tetramer with its potential substrate *SufS*, but loses its sulfotransferase activity[24]. The remaining 2 cases were genuine conflicts, caused by different publications using similar experimental approaches, but reporting different effects. A detailed overview of this comparison can be found in Supplementary Data 2.

The vast majority of the data set refers to amino acid substitutions, with a marginal amount of insertions and deletions reported (only 65 deletion and 83 insertion annotations). Figure 2a shows that arginine, leucine and serine are the most frequently replaced residues, while histidine and methionine residues are mutated less often (see Supplementary Figure 3a for a more detailed view on specific replacements). Alanine is by far the most frequently used residue for replacement (Fig. 2b), which is probably reflective of the widespread use of alanine scanning[25] to

identify residues critical for binding to other molecules, either because they are found on the interacting interface or at an allosteric binding site. When we checked the relative proportion of the different mutation effects per replacing residue (Fig. 2c, Supplementary Figure 3b), alanine replacements mostly associate with deleterious effect on interactions. The dominance of deleterious effects most probably reflects the authors of the original study using alanine scanning to locate binding-related residues.

**Genomic variation and the IMEx mutations data set**. In this era of deep-sequencing genomics, there is a wealth of data concerning nonsynonymous genomic variants. As discussed before, the motivation behind the design of these experiments varies, and only a fraction were specifically designed to systematically test known variants vs reference (wild type) versions of the participant proteins[5,26]. Hence, we decided to explore how much currently available information for natural or disease-related variation can be linked to the data set. Because of the strong predominance of human data both in IMEx mutations and in variation data sets, we decided to focus on human proteins only.

We used the EMBL-EBI Proteins API[27] to access variation data both manually annotated by and mapped to UniProtKB from large-scale sequencing studies such as the 1000 Genomes[28], ExAC[29] and COSMIC[30] projects. We queried 8820 sequence changes in 1990 human proteins, corresponding to 16,765 IMEx mutation annotations (see Table 4 and Fig. 3). 29% (4804) of the mutation annotations (Fig. 3a) and 12% (1073) of the sequence changes (Fig. 3b) were fully mapped to natural variants. We also checked cases in which there is a variant described in the same position as a mutation reported in the IMEx data set, but the amino acid change is different in the two data sets (positional matches), and also those where mutations span more than one residue and only some of the residue changes or positions are matched in UniProtKB (partial matches). Sixteen percent (2671) of the mutation annotations (Fig. 3a) and 16% (1415) of the sequence changes (Fig. 3b) are positional or partial matches. The biological significance of positional and partial mappings does not go beyond stating that the region or position in question is important for interaction and is variable. However, we believe this information might be useful for researchers interested in exploring specific regions in more detail.

We also checked how many of the mapped variant annotations have been linked to disease according to UniProtKB. Disease associations were complemented with data from the DisGeNET

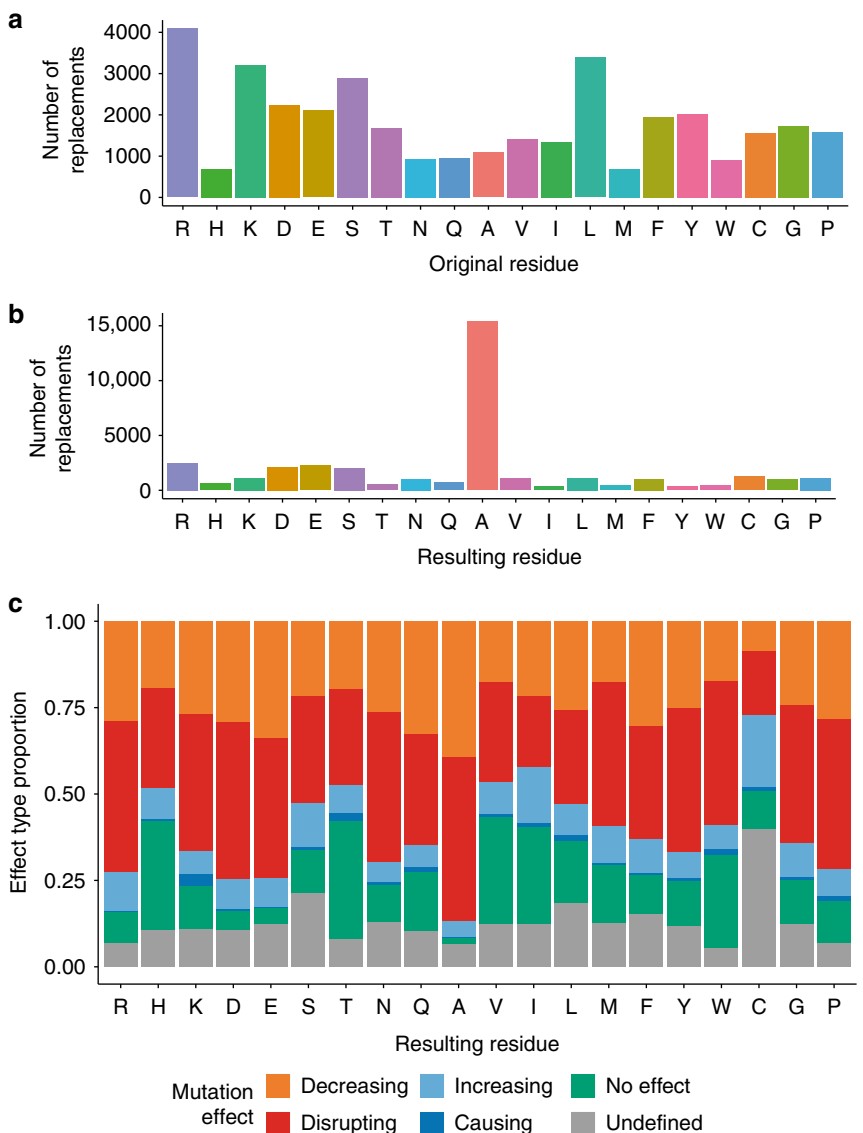

**Fig. 2** Amino acid replacement frequencies in the full data set. **a** Replacement frequencies by original residue; **b** Replacement frequencies by resulting residue; **c** Normalized frequencies of resulting sequences by mutation effect over the interaction. Substitutions with non-standard amino acids and deletions are not shown for simplicity. Source data are provided as a Source Data file

database[31]. There were disease-associated variants for 42% (840) of the proteins queried, with a median value of 4 disease variants mapped per protein. As seen in Table 4, 20% (3432) of IMEx mutation annotations have been tagged as related to disease, with over 900 known disease variants represented in the data set. UniProtKB derives disease annotations for variants from both manual curation[32] and imports of cross-referenced data from ClinVar[33] via Ensembl[34], while DisGeNET also includes variants from the GWAS Catalog[35], and from text-mining the scientific literature. Figure 3c shows the distribution of the variant associations for different disease classes using the Medical Subject Headings (MeSH)[36] classification. A detailed list of these mutation annotation-disease associations can be found in Supplementary Data 3.

We then checked if the proportion of disease-related annotations in IMEx varies depending on the reported effect on interaction. As seen in Fig. 3d, e, disease-related mutations tend to have mostly deleterious effects on interaction outcome, but we could also map a considerable number of annotations where there was an increase or even gain-of-function in terms of binding (411

annotations representing 116 variants). When we look at mutation recurrence in different types of cancer as extracted from cBioPortal[37], mutations strengthening interactions seem to have both statistically higher recurrence values and a higher proportion of mutations with extremely high recurrence in cancer data sets (Fig. 3f, g).

**Computational predictions and literature curation.** There is currently a variety of computational tools used to annotate variation data sets[38]. These tools can report the effect of variation on protein function, folding or binding, usually based almost exclusively on sequence or structural data, or can also report genome-derived parameters such as allele frequencies or conservation scores. We wanted to study how variation annotations provided by these tools align with experimental effect over interaction as reported in the literature.

For this purpose, we used mutfunc (www.mutfunc.com)[39], a database reporting the effect of almost any possible mutation on protein stability, interaction interfaces, post-translational

**Table 4 Variant mapping summary**

| Variant record type | Proteins annotated (1990 searched) | UniProtKB variants mapped to IMEx annotations | | IMEx annotations mapped to UniProtKB variants | |
|---|---|---|---|---|---|
| | | Full matches | Cumulative with partial/positional matches | Full matches | Cumulative with partial/positional matches |
| Natural variant | 1948 | 1073 | 2488 | 4804 | 7475 |
| Disease variant | 840 | 732 | 877 | 3432 | 3804 |

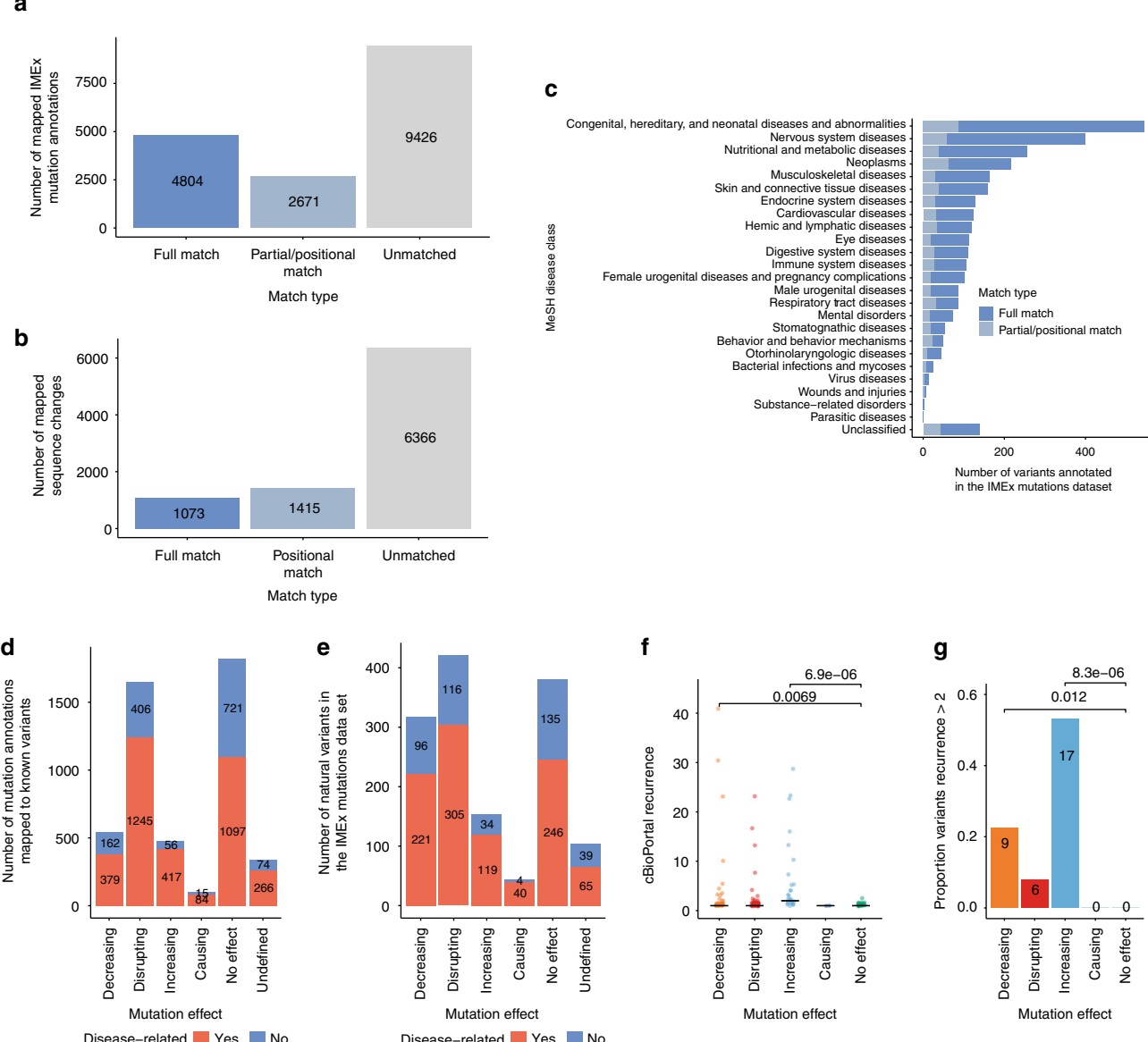

**Fig. 3** Genomic variation and disease annotations in the IMEx mutations data set. **a** Mapping IMEx mutation annotations to UniProtKB human variants; **b** Mapping UniProtKB human variants to IMEx-reported sequence changes; **c** Distribution of IMEx mutation annotations associated with selected MeSH disease classes (see Supplementary Data 3 for a complete list of disease-mutation associations). Bars are coloured according to the type of mapping between disease variants and IMEx mutation annotations (full match: blue, partial/positional match: grey); **d** IMEx mutation annotations by effect type and their relation to disease; **e** IMEx mutation sequence changes by effect type and their relation to disease; **f** cBioPortal recurrence scores for mutations grouped by effect type. $p$-values calculated with one-sided Wilcoxon test are indicated (decreasing vs no effect $W = 582.5$; increasing vs no effect $W = 521.5$); **g** Proportion of highly recurrent cancer variants according to cBioPortal by effect type. $p$-values calculated with Fisher exact test are indicated. Source data are provided as a Source Data file

modifications, protein translation, conserved regions, and regulatory regions. It hosts pre-computed variation effect data derived from established resources such as SIFT[40], Interactome3D[41] or FoldX[42].

We first examined the predicted destabilization effect of mutations on structural models of protein–protein interfaces, dividing them by the literature-reported effect. As can be seen in Fig. 4a, mutations with a 'decreasing' and especially a 'disrupting' effect on interactions had a significantly higher predicted destabilization effect than those with no effect, a difference that was not seen in mutations that would strengthen or even cause an interaction. These deleterious groups also contained a significantly higher proportion of mutations predicted to be very destabilizing for interfaces (Fig. 4b).

We next studied genome-derived parameters that are useful to study variation, such as residue conservation or natural allele frequencies. The experimentally observed impact on binding stability that we report in our data set may also be reflected on these parameters. This assumption was partially confirmed using three independent measurements. Firstly, we used the 'sorting intolerant from tolerant' (SIFT) method[40], observing that the proportion of variants with low tolerance scores was significantly higher in all groups where an effect was reported vs the 'no effect' reference (Fig. 4c). Secondly, we also checked allele frequencies as derived from ExAC data. Again, mutations with a reported effect seemed to have significantly lower allele frequencies (Fig. 4d) and a higher proportion of alleles with extremely low frequencies (Fig. 4e) than those reported to have no effect over interaction. Finally, we used PolyPhen2[43], a missense mutation effect prediction algorithm that uses a naive Bayes classifier based on both sequence-based and structure-based predictive features. In this case, mutations predicted to be deleterious by PolyPhen2 were significantly enriched in the 'decreasing' and 'disrupting' effect groups vs those with 'no effect' (Fig. 4f).

The interaction-perturbing effects reported in the IMEx data set can be caused by modifying overall protein structure or by alteration of binding interfaces. We can determine if the mutations reported fall within sequence regions associated with binding using both computational predictions and literature-reported experimental data. We obtained predicted interfaces, based on available structural data, from Interactome3D[41]. Literature-curated interfaces were inferred from IMEx records that contained participant features of the 'binding-associated region' (MI:0117) branch. These represent experiments where the authors have tested fragment constructs in an attempt to find sequence regions that are critical for binding, although they may not necessarily represent the actual binding surface. As seen in Fig. 4g, h, most of the mutations fall within predicted or curated interfaces. The proportion of mutations having an effect over the interaction seems to be higher in binding interfaces, both predicted and inferred from IMEx curation. Disease-associated variants seem to show the same pattern (Supplementary Figure 4a, b). Thus, the majority of the variants reported to have effects on protein interactions (68%) can be linked to perturbations inside binding regions, with a smaller proportion of variants (32%) potentially representing systemic or allosteric effects influencing interactions.

**Phosphorylation and mutations in interactions**. Post-translational modifications (PTMs) can be regarded as chemical switches with the potential to influence protein interactions[44]. We explored whether mutations with reported effects in our data set are enriched in known sites of protein modification, focusing on phosphorylation in human proteins as the best characterized PTM data set. Using phosphosite annotations from PhosphoSitePlus®[45], we found that decreasing, disrupting and increasing mutations are indeed enriched in annotated phosphosites (Fig. 5a). Specific amino acid substitutions are commonly used to either disrupt a phosphosite (phospho-disrupting mutations, where serine, threonine or tyrosine are replaced with alanine, glycine, valine or phenylalanine) or to simulate it (phospho-mimetic mutations, with the same amino acids replaced by glutamic or aspartic acid). Looking at these substitutions specifically, we can see that phosphosite-disrupting mutations account for the majority of the overlap with annotated phosphosites and are again enriched for decreasing, disrupting and increasing effects (Fig. 5b). Phospho-mimetic mutations are significantly enriched only when they have an increasing effect over interaction (Fig. 5c).

**Literature bias in the IMEx mutations data set**. IMEx databases have a wide scope when selecting publications for curation and it is reasonable to assume that the proteins in this data set are representative of the interaction data that has been explored in the literature. Socially driven, literature bias is a well-known phenomenon previously reported for literature-curated data sets[26,46], so we decided to explore to what extent it affects the data set.

First, we checked whether the number of annotations and variants found in the data set and the number of publications in which the affected protein is reported are correlated. As seen in Fig. 6a, b, the data set contains examples of both heavily researched proteins with a low number of annotations and variants and vice versa. If we fit linear models between the number of annotations/variants and number of publications in which a protein is reported we find a slight positive correlation, especially in the case of disease-related variants. This observation is compatible with socially-induced bias, with known disease-related proteins and variants being more often reported in the literature.

Then we set out to find if the proteins represented in the mutations data set are involved in distinctive pathways vs all proteins for which IMEx has interaction information. To avoid database-specific biases we performed annotation enrichment analysis using PathDIP (http://ophid.utoronto.ca/pathDIP), an analysis tool that integrates information from 20 source databases[47]. Human proteins were divided in different sets depending on the effect reported for their mutations and their pathway annotation enrichment was calculated using all the human proteins in IMEx as background. Pathways obtained from these sets have substantial overlap (Fig. 6c, 694 pathways). These results suggest that the proteins whose mutation effect on interactions have been collected in this data set may be biased, possibly due to specific interest of the researchers exploring variation influence on molecular interactions. Specifically, in the group of mutations that show an effect on interactions, pathways related to the immune system, signalling, disease and cell cycle control ranked on the top (Supplementary Figure 5, see Supplementary Data 4 and 5 for full details), with little difference between effect categories. There seems to be a predominance of cancer-related pathways, with representatives in both the 'disease' and the 'signalling' categories, which agrees with the observation reported in Fig. 6b that the literature is biased towards disease-related variants.

## Discussion

Here we present a unique resource containing experimental, publicly available information about the impact of sequence changes on specific protein–protein interaction outcomes. This is a direct result of the IMEx Consortium full-detail curation policies and represents an example of how expert curation, resulting in structured and standardized representations, is required in

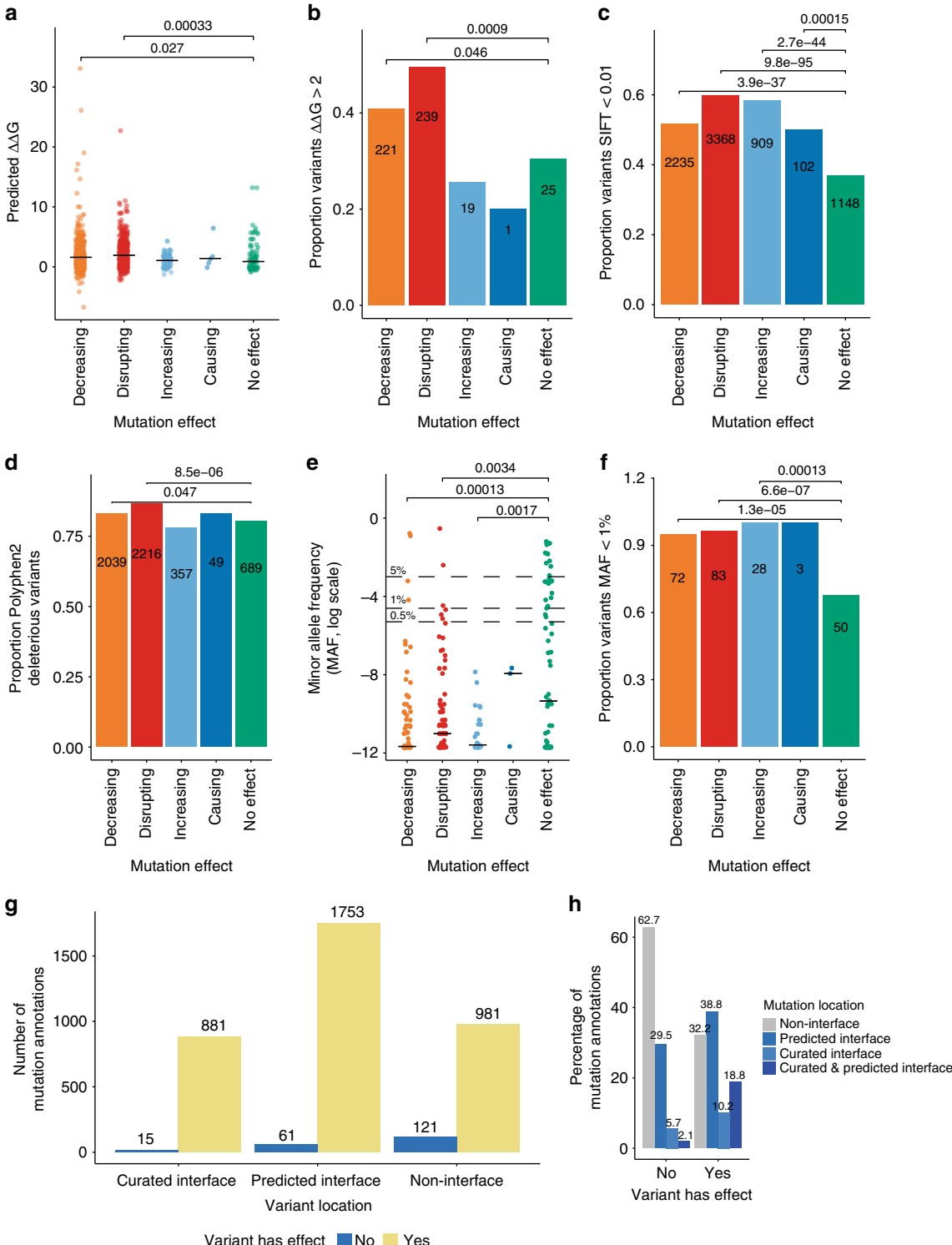

**Fig. 4** Computational annotations and the IMEx mutations data set. **a** Interaction interface disruption as predicted with FoldX, by mutation effect type. Indicated *p*-values calculated with Wilcoxon test (decreasing vs no effect $W = 25{,}100$; disrupting vs no effect $W = 24{,}411$); **b** Proportion of highly disruptive variants by mutation effect type; **c** Proportion of low tolerance residue positions according to the SIFT, by mutation effect type; **d** ExAC-extracted allele frequencies for mutations represented in the IMEx data set, by mutation effect type; **e** Low frequency variants, by mutation effect type. Indicated *p*-values calculated with Wilcoxon test (decreasing vs no effect $W = 1841.5$; disrupting vs no effect $W = 2389.5$; increasing vs no effect $W = 645$); **f** Proportion of deleterious substitutions according to Polyphen2, by mutation effect type; **g** Number of mutation annotations located in binding interfaces (curated and predicted), by effect; **h** Normalized frequencies of mutation annotations reporting effects over interactions or not and their localization in binding interfaces. *p*-values indicated in panels **b**, **c**, **d** and **f** were calculated with Fisher exact test. Source data are provided as a Source Data file

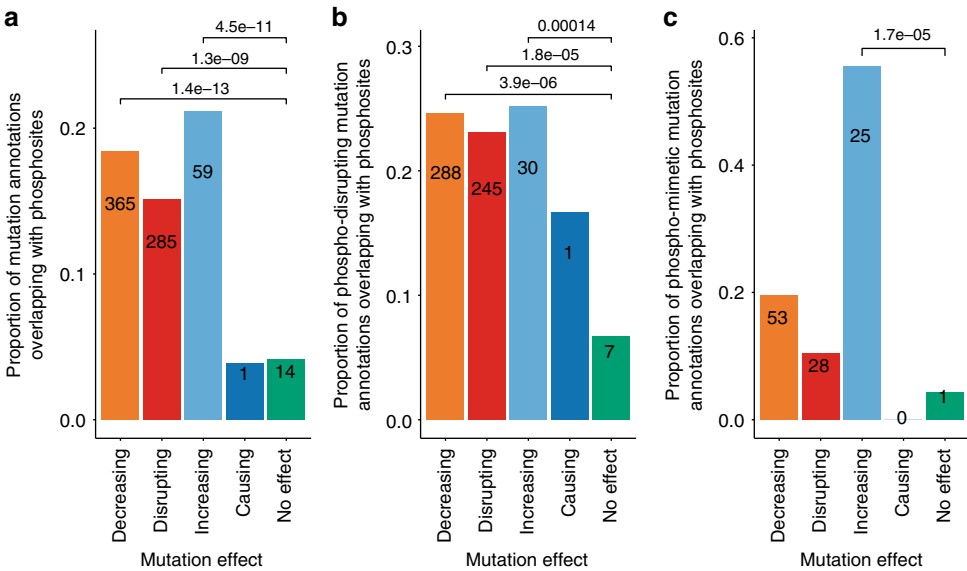

**Fig. 5** Overlap between phosphosites and mutation annotations. Overlap of mutation annotations with human phosphosites, as taken from PhosphoSitePlus®. **a** Overlap with all mutation annotations in all potential phosphosites; **b** Overlap with phospho-disrupting mutation annotations; **c** Overlap with phospho-mimetic mutation annotations. *p*-values for comparison with the reference 'No effect' effect type indicated in the figure were calculated with Fisher exact test. Source data are provided as a Source Data file

order to make the most of published experimental results. In comparison to similar, pre-existing data sets recording variation influence over interactions, this resource represents a leap forward in depth, size and scope (Table 3). A previous, relatively small study[48] reported a curated list of about 100 mutations influencing interactions. Despite obvious limitations due to its size, this was used as benchmark in a study investigating the link between disease-related variation and interaction interfaces[1], showing the applied potential of this type of data. The curation infrastructure and practices of the IMEx Consortium will enable the capture of data from a growing number of deep-mutagenesis interaction studies, where hundreds if not thousands of single amino acid changes over the whole length of a protein sequence are explored for their influence on interactions[49]. New studies are appearing in which the complex relationship between genetic and protein interactions is explored[50]. These epistatic relations can potentially be represented in the PSI-MI data model and the curators in the IMEx Consortium are exploring ways to enable their annotation.

We have also acknowledged the social biases inherent to any literature-based resource in our data set, although it is difficult to ascertain its extent. Alanine scanning features prominently as a commonly used technique (Fig. 2b) and may represent amino acid changes that will never be seen in nature due to evolutionary constraints or simply because they would require extensive sequence alteration at the DNA level, but remains an invaluable source of information, identifying key binding-related positions. For the human sub-section of the data set, disease-related variants and proteins are possibly over-represented (Fig. 3b, c, Supplementary Figure 5) and have been preferentially selected for biocuration over non-disease-related proteins (Supplementary Figure 2a). Interestingly, we report over 100 disease-related variants described in the literature to either cause or increase existing interactions (Fig. 3d, e), some of which are found to be highly recurrent in cancer according to cBioPortal[51]. This contrasts with the findings reported by Sahni et al.[5], where only two cases of gain-of-interaction mutations were found in a systematic screening for disease-related mutations and their effect on interactions using yeast two-hybrid technology. Although

interaction decreasing/disrupting effects were much more frequently reported, this highlights how gain-of-interaction mechanisms could play a significant role in disease pathogenesis, especially in cancer.

We have also explored mutations that mimic or disrupt known phosphorylation sites, finding enrichment in mutations with reported effects. Phosphosite-disrupting mutations predominantly show detrimental effects on protein interactions, and phosphosite-mimetic mutations have both detrimental and increasing effects (Fig. 5). These results are consistent with phosphorylation being regarded as a molecular switch influencing binding function.

Analysis of variation is a fundamental tool in basic and clinical research, with direct application in the clinic through translational genomics. Variation effects are explored mainly through statistical analysis of large population data sets, GWAS studies, or by quantitative analysis of its influence on expression via identification of eQTLs. However, in order to unravel the mechanisms behind detected effects, it is key to explore how molecular interactions are affected[52]. Currently, most of the mechanistic insight into variation effects is generated by computational annotation and predictions, using tools that are based on relatively small reference sets, generally based on structural data. As an example, the widely used FoldX algorithm is generated from protein complex structures and has been tested against a library of 1008 mutants[42]. Our current data set already provides interaction effects for over 10 times more individual variants and is not limited to structural data. The wide scope of experimental setups represented (Fig. 1c) allows the capture of effects on proteins and protein regions that might be intrinsically non-structured[53]. We show that the data set gives a currently unparalleled and representative overview about which residues are key for protein interactions, with the results being in good accordance with commonly used variant annotators (Fig. 4). IMEx curation practices originally did not enforce capturing sequence changes that had no effect over interaction outcome, but as a result of consultations with tool developers and data users this policy has been amended and the data set now features a growing number of mutations with no effect that can be used as a

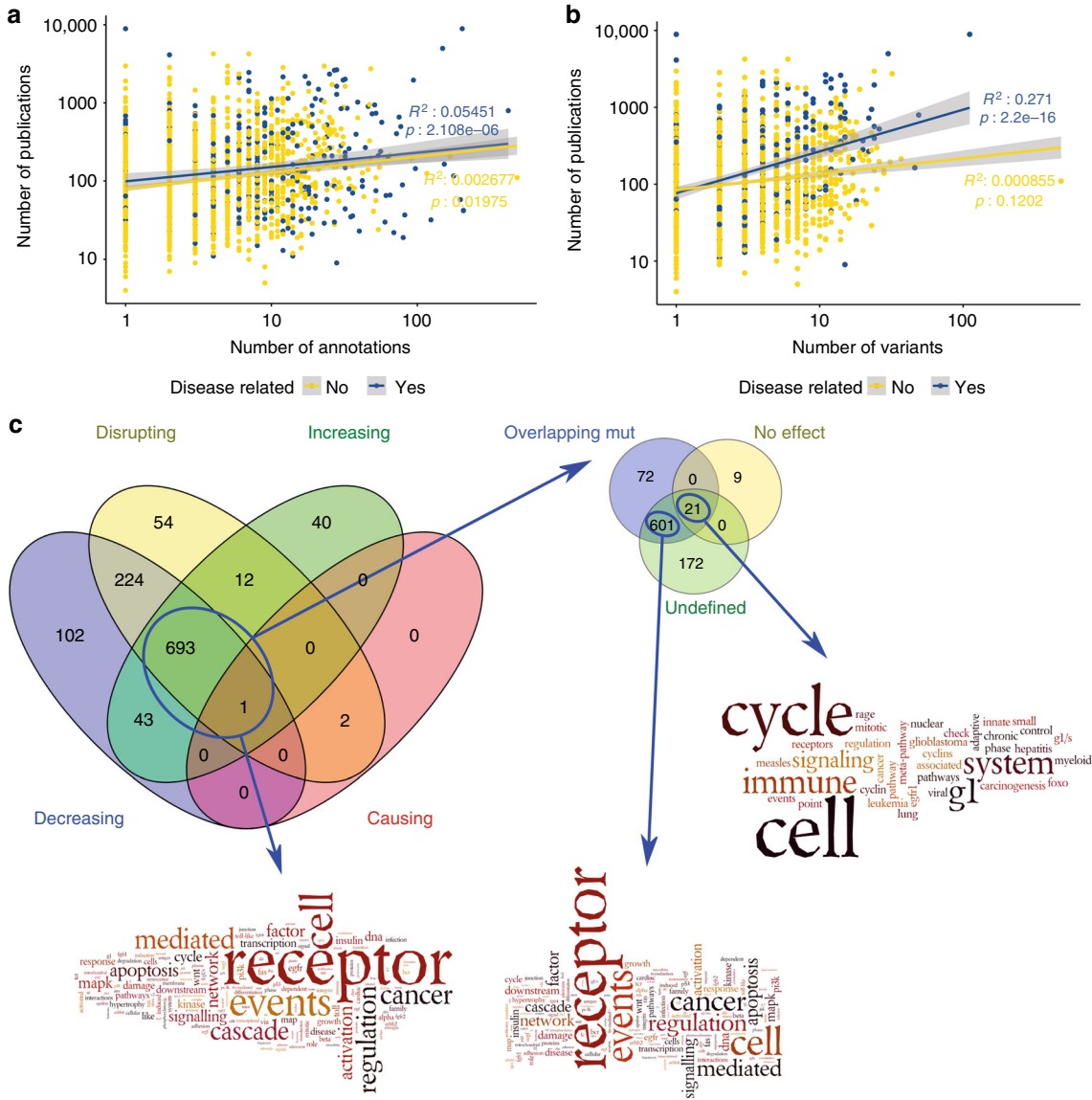

**Fig. 6** Literature biases in IMEx mutations data set. **a**, **b**: Scatter plot of number of publications (in a logarithmic scale) in which a protein is reported vs **a** the number of annotations and **b** the number of variants reported in the IMEx mutations data set. Disease-related annotations are highlighted in blue, unrelated to disease in yellow. Regression lines are coloured accordingly. Correlation coefficients and p-values for the linear model fit are indicated. 95% Confidence Interval (CI) is shown as greyed areas over the regression lines; **c** Overlap of significantly enriched pathways (q < 0.01) across different sets of proteins and word enrichment analysis (using Wordle on enriched pathway names) for the overlapping set (any mutational effect), the set of proteins annotated with no effect and the remaining proteins in IMEx (non-mutated). In 'No effect' word enrichment analysis, the words 'pathway' and 'action' have been removed to make remaining words more visible (original word cloud available as Supplementary Figure 6a), while in 'Overlapping mut' word cloud the words 'pathway' and 'signalling' have been removed (original word cloud available as Supplementary Figure 6b). The analysis in this figure was performed considering human proteins only. Source data for Fig. 6a and 6b are provided as a Source Data file, source data for Fig. 6c is available as Supplementary Data 4 and 5

training negative set for the development of computational annotation tools.

The IMEx mutations data set represents both a reference source for direct, literature-based variant characterization and a unique benchmark that can be used to further refine computational variant effect annotators. We will continue to expand the data set and improve its accessibility for users, as a part of IMEx global mission of ensuring data representation and re-use.

## Methods
**Source data**. All analysis was performed using the September 2017 version of the IMEx mutations data set, which can be directly downloaded from ftp://ftp.ebi.ac. uk/pub/databases/intact/2017-09-02/various/mutations.tsv.

**Software and packages used**. The quality control pipeline for mutation annotations was developed and integrated within the production code used in the IntAct database. The code is written in Java and makes use of the Hibernate and Spring frameworks for interaction with the core SQL database and application implementation. Specific implementation details are available upon request. Statistical analysis, plots, mutation re-annotation checks and mappings were performed using the R programming language[54] through the RStudio programming suite[55]. The following R packages were used in the study: data.table, dplyr, ggplot2, ggpubr, gridExtra, gsubfn, httr, jsonlite, plyr, RCurl, reshape2, scales, seqinr, splitstackshape, XML, Biostrings, and biomaRt.

**Curation practices**. Data has been produced through manual literature curation following the IMEx Consortium curation guidelines[11], which can be explored in detail on the Consortium's website: www.imexconsortium.org/curation. Briefly, every publication reviewed was curated for the entirety of the interaction data it

contained, representing each experimental piece of evidence as a separate record. Full details of constructs used were registered and every entry was reviewed by at least two independent curators for quality control.

**Mutations re-annotation effort.** After the development of PSI-XML3.0 and the 'resulting sequence' field in the IMEx schemas to capture amino acid change in participant features of the type 'mutation (MI:0118)' and children, it was necessary to populate the field with legacy data from the participant feature short label. This free-text, manually-entered field was prone to contain typographical errors and was difficult to keep updated. Curators used a set of simple rules to depict amino acid substitution, deletions and insertions. As a first step towards populating the 'resulting sequence' field, we wrote ad hoc parsing scripts to evaluate and extract the information stored in the short labels. Several rounds of corrections took place until the data set got to its current state. Of the 27,868 records of the data set, 20,161 had to be corrected, with around 2000 of them manually corrected. There are still about 2500 records for which no fix was possible without fully amending the original entry. These have been left out of the data set until being revisited by an IMEx curator in due course. An automated quality control pipeline has been put in place to handle newly-created entries and future changes in UniProtKB (details in Supplementary Information section 'Initial re-curation of mutation data in IMEx'). Finally, we have also adapted the participant feature short labels to the Human Genome Variation Society (HGVS) recommendations for variant annotation[56], which can be accessed at http://varnomen.hgvs.org/recommendations/protein/.

**IMEx, UniProt and SKEMPI 2.0 mutation annotations comparison.** The latest version of the SKEMPI 2.0 database was downloaded in July 2018 from https://life.bsc.es/pid/skempi2/database/download/. The entries were parsed and the PDB REST API was used to map the proteins in the PDB records to UniProt accession numbers in order to enable comparison. Chain information was used to ascertain the identity of the individual proteins. Cases where more than one protein was reported by chain, as well as unmapped entries, were discarded from further analysis. Finally, effect types were classified in two categories: 'loss' and 'gain' of binding function. In order to classify the annotation in one of these two categories, the affinity values for the mutant version of the sequence were normalized in relation with the wild type form. Fold changes over and under 50% with respect of the wild type were annotated as 'gain' or 'loss', respectively. Smaller fold changes were qualified as 'no effect'. UniProt mutagenesis annotations were obtained on 29 July 2018 parsing the UniProt webservice XML output using a Python script kindly provided by Luz García-Alonso. Since interacting partners are only identified with gene symbols in UniProt annotations, their precise identity could not be ascertained for a number of records, which were discarded for further analysis. The effects reported in the annotations were simplified and qualified as simple 'loss' and 'gain' of binding function. IMEx mutation types were simplified accordingly to allow consistent comparison of mutation effects among data sets.

**Assessment of conflicting mutation annotations.** A mutation was defined as 'conflicting' when a sequence variant, tested against the same interacting partners, was annotated for effects that are directly antagonistic (e.g. 'disrupting' vs 'increasing'). Different proteoforms of the same reference protein were considered as different interacting partners (e.g. a phosphorylated protein was considered different from its non-phosphorylated form). When different effects were reported but were not directly antagonistic the mutation was considered to be 'consistent' if all effects went in the same direction (e.g. 'disrupting' and 'decreasing') or 'mild conflict' if most went in the same direction and the rest were annotated as having 'no effect' (e.g. 'decreasing' and 'no effect').

**Mapping IMEx mutations to UniProtKB and the genome.** UniProtKB accessions for human proteins were extracted from the IMEx mutations data set, retaining isoform identifiers, and used to query the EMBL-EBI Protein API[27]. The API's 'variation' method was used to extract large-scale variation annotation from UniProtKB, regardless of its origin. Annotations extracted through this method were then mapped to the IMEx mutations data set using UniProtKB accession, sequence position and resulting amino acid for 'full' mappings and only UniProtKB accession and position for 'positional' mappings. Cases where the IMEx-reported mutation spans more than one amino acid position were split into individual substitutions and only labelled as 'full' matches if every individual position matches an annotation in UniProtKB. Otherwise, they were considered 'partial' mappings. Disease annotations were extracted from the API's output, along with rsIDs. These rsIDs were then used in DisGeNET to search for additional disease annotations that were brought in as well. The diseases were mapped to their corresponding MeSH disease classes using the cross-references provided by the Unified Medical Language System (UMLS) Metathesaurus (version 2017AB). In the cases in which cross-references to MeSH were not available, we manually assigned a parent, for example, Sporadic Breast Carcinoma was mapped to Breast Carcinoma. This allowed to map 98.5 and 93% of the diseases in UniProt and DisGeNET, respectively. The following MeSH disease classes were discarded because they were deemed uninformative or not appropriate for the analysis in

hand: 'pathological conditions, signs and symptoms', 'occupational diseases' and 'animal diseases'.

**Predicting impact on protein interaction interfaces.** Experimental and homology modelled structures for protein interactions were obtained from the Interactome3D database[41]. Relative solvent accessibility (RSA) for all residue atoms was computed using NACCESS[57] for proteins individually and in the interaction complex. Interface residues were defined as those with any change in RSA. The impact of a variant on interface stability was computed using FoldX v.4.0. All binary interface structures were repaired using the RepairPDB command, with default parameters. The Pssm command was then used to predict ΔG with numberOfRuns = 5. This performed the mutation multiple times with variable rotamer configurations, to ensure the algorithm achieved convergence. The average ΔG of all runs was computed and the ΔΔG was computed as the difference between the wild type and mutant, providing a predictive estimate of how destabilising the mutant was to the interaction interface.

**Predicting variant functional impact with SIFT and PolyPhen2.** In order to calculate SIFT scores, all protein alignments were built against UniRef50[58], using the seqs_chosen_via_median_info.csh script in SIFT 5.1.1[59]. The siftr R package (https://github.com/omarwagih/siftr) was used to generate SIFT scores with parameters ic_thresh = 3.25 and residue_thresh = 2. PolyPhen2 scores were obtained from http://genetics.bwh.harvard.edu/pph2/bgi.shtml using the batch query tool on 6 August 2018. The GRCh37/hg19 genome assembly and the HumDiv classifier model were used for considering missense annotations against canonical transcripts only.

**Allele frequencies.** A total of 3,198,692 coding variants in *H. sapiens* for over 65,000 individuals was collected from the ExAC Consortium[29] in the ANNO-VAR[60] output format along with corresponding adjusted allele frequencies. Ensembl transcript positions were mapped to UniProt by performing Needleman-Wunsch global alignment of translated Ensembl transcript sequences against the UniProt sequence using the pairwiseAlignment function in the Biostrings R package. The mapping between Ensembl transcript IDs (v81) and UniProt accessions was obtained from the biomaRt R package. In the case that multiple alleles mapped to the sample single amino acid substitution, the one with the highest adjusted allele frequency was retained.

**Recurrence.** Annotated somatic mutation recurrence data for 10,155 tumour samples was obtained from the TCGA pan-cancer atlas data set downloaded from cBioPortal v1.15.0 on 11/08/2018. The data set comprises 1,866,976 missense variants mapped to UniProt and belonging to 33 tumour types.

**Mapping variants to interaction interfaces.** Predicted interface and accessibility coordinates were obtained from Interactome3D. Curated interfaces were extracted from IntAct by selecting participant features under the PSI-MI term 'binding-associated region' (MI:0117). Only human proteins for which accessibilities were calculated directly from structural data in Interactome3D were selected for this analysis, modelled structures were excluded.

**Mapping mutations and phosphosites.** Phosphosite data was downloaded from PhosphositePlus® (www.phosphosite.org) on 02/07/2018. All phosphosite annotations for human proteins were considered for the analysis and mapped to our data set using provided UniProt coordinates. Only mutations on serine, threonine or tyrosine residues were used for the comparisons.

**Estimating literature bias.** We used the NCBI 'geneID2pubmed' table, accessible at ftp://ftp.ncbi.nih.gov/gene/DATA/gene2pubmed.gz, to estimate how many papers were associated to individual proteins in the IMEx mutations data set. Only human proteins were considered. Entrez GeneIDs were mapped to UniProtKB accessions using UniProt's website REST API mapping service as described at www.uniprot.org/help/api_idmapping.

**Pathway enrichment analysis using PathDIP.** Pathway enrichment was performed using mutated PPIs (i.e. mutated protein + partner) of a given mutation type (causing, disrupting, etc.) and pathDIP 2.5 pathways (considering only core pathway, http://ophid.utoronto.ca/pathDIP[47]). We considered whole IntAct human PPIs as a background for enrichment analysis (downloaded 24 March 2018). For pathways overlap Venny 2.1.0 (http://bioinfogp.cnb.csic.es/tools/venny) was used and Wordle (www.wordle.net) was used to prepare word clouds from enriched pathway titles.

**Code availability.** The code and mappings used for analysis are available upon request.

## Data availability

The IMEx mutations data set is open access and fully available at www.ebi.ac.uk/intact/resources/datasets#mutationDs under an open Creative Commons Attribution 4.0 International licence (CC-BY4.0), like all IMEx data. All additional data used in this publication is open access and sources are cited where appropriate.

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

## Acknowledgements

The IntAct database and EMBL-EBI-based authors received funding from EMBL core funding and Open Targets (grant agreement OTAR-044). The DIP database is funded by NIH grant R01GM123126. MINT received support from ERC grant 'DEPTH project of the European Research Council (grant agreement 322749)'. The British Heart Foundation-University College of London (BHF-UCL) curation team is funded with the British Heart Foundation grant RG/13/5/30112. UniProt curation activities at EMBL-EBI and the Swiss Institute of Bioinformatics are funded by the National Eye Institute, National Human Genome Research Institute, National Heart, Lung, and Blood Institute, National Institute of Allergy and Infectious Diseases, National Institute of Diabetes and Digestive and Kidney Diseases, National Institute of General Medical Sciences, and National Institute of Mental Health of the National Institutes of Health under Award Number [U24HG007822], National Human Genome Research Institute under Award Numbers [U41HG007822 and U41HG002273], and the National Institute of General Medical Sciences under Award Numbers [R01GM080646, P20GM103446 and U01GM120953] (the content is solely the responsibility of the authors and does not necessarily represent the official views of the National Institutes of Health); Swiss Federal Government through the State Secretariat for Education, Research and Innovation; and aforementioned British Heart Foundation grants and EMBL core funding. DisGeNET is supported with EU-FP7 funds from ISCIII-FEDER (CP10/00524, CPII16/00026), IMI-JU (grant agreement no. 116030, TransQST) and EFPIA companies in kind contribution, and the EU H2020 Programme 2014-2020 (grant agreements no. 634143, MedBioinformatics and no. 676559, Elixir-Excelerate). The Research Programme on Biomedical Informatics (GRIB) is a member of the Spanish National Bioinformatics Institute (INB), PRB2-ISCIII and is supported by grant PT13/0001/0023, of the PE I+D+i 2013-2016, funded by ISCIII and FEDER. The DCEXS is a 'Unidad de Excelencia María de Maeztu', funded by the MINECO (ref: MDM-2014-0370). I.J. and group supported in part by Krembil Foundation, Ontario Research Fund (#34876), and Canada Foundation for Innovation (CFI #225404, #30865, #33536). The authors would like to thank Marco Galardini, Luz García-Alonso, Denes Turei and Martin Krallinger for valuable discussions when designing the data set output format; Iain Moal for providing key information about SKEMPI 2.0; Danish Memon for his help pre-processing cBioPortal data; and Luz García-Alonso as the creator of the Python scripts we used to parse UniProt mutagenesis annotations.

## Author contributions

S.O. and P.P. designed this study and wrote the manuscript. The IMEx Consortium curators generated the mutation annotations. M.D., S.O., M.Koch, N.dT., A.S. and P.P. re-curated the data set and implemented semi-automated quality control procedures. L.P., D.O., O.W., C.P., M.Kotlyar, J.P. and P.P. analysed the data. S.O., H.H., P.B., L.F., I.J. and P.P. interpreted the results and revised the manuscript.

## Additional information

**Competing interests:** The authors declare no competing interests.

## IMEx Consortium contributing authors

IntAct team: J. Khadake[1], B. Meldal[1], S. Panni[8], D. Thorneycroft[1], K. van Roey[1], DIP team: S. Abbani[9], L. Salwinski[9], M. Pellegrini[9], MINT team: M. Iannuccelli[10], L. Licata[10], G. Cesareni[10], SwissProt-UniProt team: B. Roechert[11], A. Bridge[11], HPIDb team: M.G. Ammari[12], F. McCarthy[12], I2D team: F. Broackes-Carter[5], BHF-UCL team: N.H. Campbell[13], A.N. Melidoni[1,13], M. Rodriguez-Lopez[1,13], R.C. Lovering[13], MBInfo team: S. Jagannathan[14], InnateDB team: C. Chen[15], D.J. Lynn[16,17], MatrixDB team: S. Ricard-Blum[18], Molecular Connections team: U. Mahadevan[19] & A. Raghunath[19]

[8]DiBEST Department, University of Calabria, Via Pietro Bucci, Rende (CS) 87036, Italy. [9]Los Angeles Department of Energy Institute for Genomics and Proteomics, University of California, Los Angeles, CA 90001, USA. [10]Department of Biology, University of Rome Tor Vergata, Via della Ricerca Scientifica, 00118 Rome, Italy. [11]Swiss-Prot group, SIB Swiss Institute of Bioinformatics, CMU 1, Rue Michel Servet, 1211 Geneva 4, Switzerland. [12]School of Animal and Comparative Biomedical Sciences, University of Arizona, Tucson, AZ 85721, USA. [13]Functional Gene Annotation, Institute of Cardiovascular Science, University College London, London 56273, UK. [14]Mechanobiology Institute, National University of Singapore, T-Lab #05-01, 5A Engineering Drive 1, Singapore 117411, Singapore. [15]Centre for Microbial Diseases and Immunity Research, University of British Columbia, Lower Mall, Vancouver 2259, Canada. [16]South Australian Health and Medical Research Institute, EMBL Australia Group, North Terrace, Adelaide, SA 5001, Australia. [17]The College of Medicine and Public Health, Flinders University, Bedford Park, SA 5042, Australia. [18]Univ. Lyon, ICBMS, UMR 5246 CNRS, Université Lyon 1, Villeurbanne Cedex, 69622 Lyon, France. [19]Molecular Connections Pvt. Ltd., Kandala Mansions, 2/2 Kariappa Road, Basavangudi, Bangalore 560004, India

