## [Peer Review File · Nature Communications]

Reviewers' comments:

Reviewer #1 (Remarks to the Author):

The database presented is the largest collection of mutations experimentally tested for their impact on protein-protein interactions. This dataset is then systematically analyzed to give useful summary statistics, and compared against a variety of amino acid resolution datasets to both validate the functional relevance of the collection, and to provide disease context. This dataset is timely given the recent advances in deep scanning mutagenesis technologies that will lead to an explosion of this type of data in the coming years. It also represents a huge amount of curation work and should be of great value for the protein interaction community and beyond. The paper is well written and easy to understand, analyses appear sound and the figures simple to interpret.

Analytical points:

1) The only large scale amino acid resolution dataset that is not used for a functional comparison is that of post-translational modifications (PTMs), many of which are known to impact protein-protein interactions (as the authors note in their introduction). There are >100,000 PTMs recorded in the online phosphositeplus database that would provide an interesting comparison with sub-categories of mutation detected here, and highlight the relevance of the dataset to wider communities. From Fig 2a there looks to be >10,000 modifiable residues in their dataset (methylation, acetylation, ubiquitination and phosphorylation on R,K,S,T and Y) which should provide enough data to do simple, quick comparisons. For example, are certain PTM modified residues enriched or depleted in certain mutation sub-types (disrupting / increasing etc) compared to their non-modified counterparts. More specifically, from supplementary Figure 3d we can see that when serine is mutated to aspartic acid, we see a higher normalized frequency of "increasing" interactions. Are these sites more often phosphorylated than we would expect by random chance?

2) With regards to contradictory results, the authors quote: "One reason for these contradictory results may be differences in experimental methodologies used to measure the effect, since IMEx databases recognize a large variety of experimental approaches that provide molecular interaction evidence."

Do they not have the data to test this? At least on a fundamental principle of assay type. So do they come from immuno-precipitations versus two-hybrid. Or are the experimental differences more subtle?

Discussion points:

With more of an eye on the future, can the authors comment on how they are going to deal with deep scanning mutagenesis results, which they briefly allude to in the discussion. Specifically two points come to mind:

1) Will the database have standards for defining which mutants were tested in DMS approaches, as this could very strongly impact the "tested but no effect" category? In principle 10,000s of mutants can be synthesized en masse, but that doesn't mean all are actually present in the libraries. Furthermore what is sequenced in the library input depends strongly on the sequencing error rate and number of reads. Will they leave it to each research group to define their mutant library coverage or set sequencing standards before the data can be submitted?

2) The first comprehensive mutagenesis on both sides of a protein interaction has been undertaken to define both individual mutation impact and epistasis between mutations (Diss, Elife, 2018). It surely will not be the last. Can the authors say how / whether this database will deal with type of data (epistasis)?

Minor comment:

X axis labels on figures 3d and 3e need shifting to the left.

Reviewer #2 (Remarks to the Author):

The authors provided an open access resource created by IMEx database curators for describing the effects of mutations on protein interactions. The topic is very interesting, however, the manuscript just offers a descriptive analysis of the data set. It lacks deep biological knowledge and it is difficult to find how to use this dataset for functional investigation of the mutations in complex diseases. It may be more suitable for a more specialized database journal such as bioinformatics or Nucleic acids research.

1. The authors compared IntAct with UniProtKB and SKEMPI, but it is unknown how many of the curated records are consistent among different databases.
2. It is not clear how many of these mutations are disease/cancer-related mutations, it would be better for the authors to provide the disease types.
3. It is unknown whether these mutations are identified from low throughput experiments or computational predictions. The authors should compare whether the mutations from different sources will influence the conclusions.
4. The authors only used SIFT to evaluate the deleterious mutations, there are also many other tools and the accuracy of these tools are different. Other tools need to be compared to test the accuracy and the overall predictive value.
5. It is known that the majority of the protein interaction networks are biased to well-known proteins. However, it is unknown whether the curated data is biased for these proteins.
6. Are there any rules that can be learned from these curated mutations for developing computational methods to predict which mutations are more likely to perturb PPIs?
7. The mutations curated from different literature sources may be from different genome versions, how do the authors deal with this issue? Are the conclusions affected by this issue?
8. There are different types of mutations, including point mutations, insertion or deletion, but it is difficult to know which mutations were used in each section. The authors should classify the different mutation types clearly in the manuscript.
9. The authors used a old version of the cBioPortal data. This version of the data is out of date since this database is updated by MSKCC quite frequently. The authors should check to see if any of the conclusions change significantly with these new data.

Reviewer #3 (Remarks to the Author):

In this paper the authors present the result of a titanic data curation effort to capture the impact of gene variation on protein-protein interactions. The final result is a database containing about 28,000 gene variations involving almost 14,000 protein interactions. The data is indeed very interesting, and it is worth being published, either in Nat Commun or somewhere else. Having said that, the manuscript is quite tedious, with long paragraphs describing the annotation process which, although very necessary, does not bring any new information. Additionally, from a more scientific perspective, the novelty of the manuscript is quite limited, since the authors apply a battery of very standard computational methods to the included variations (structure prediction, foldX, etc). At the end of the day, the value of the paper is the curated data, and I think it would be much clearer if the authors would just plainly describe the database, its contents and organization.

I am a bit puzzled by the results presented. The paper reports that, according to their annotations, only 11% of the aminoacid variations show no effect on the interaction patterns of the corresponding proteins. However, this figure is significantly higher in some of the main papers used to obtain and curate the data. For instance, Sadhi and co-workers (Cell 2015), which reported the effect of about 4,000 variations on the interaction profiles, saw that disease-causing mutations (i.e. more severe) cause little or no effect on 43% of the studied proteins. And this number goes up to almost 100% for non-disease mutations. Similar trends can also be derived from Yang and colleagues (Cell 2016), where they studied the interaction patterns of splice variants (i.e. protein isoforms). Obviously, they often involve much drastic changes than just a residue and, nevertheless, about 60% of the isoforms of the proteins studied only changed marginally (less than 50%) their interaction patterns. I am thus surprised of the large impact on the interaction profiles reported in this paper. If they are indeed correct, the authors should discuss these apparent discrepancies.

Reviewers' comments:

Reviewer #1 (Remarks to the Author):

The database presented is the largest collection of mutations experimentally tested for their impact on protein-protein interactions. This dataset is then systematically analyzed to give useful summary statistics, and compared against a variety of amino acid resolution datasets to both validate the functional relevance of the collection, and to provide disease context. This dataset is timely given the recent advances in deep scanning mutagenesis technologies that will lead to an explosion of this type of data in the coming years. It also represents a huge amount of curation work and should be of great value for the protein interaction community and beyond. The paper is well written and easy to understand, analyses appear sound and the figures simple to interpret.

Analytical points:

1) The only large scale amino acid resolution dataset that is not used for a functional comparison is that of post-translational modifications (PTMs), many of which are known to impact protein-protein interactions (as the authors note in their introduction). There are >100,000 PTMs recorded in the online phosphositeplus database that would provide an interesting comparison with sub-categories of mutation detected here, and highlight the relevance of the dataset to wider communities. From Fig 2a there looks to be >10,000 modifiable residues in their dataset (methylation, acetylation, ubiquitination and phosphorylation on R,K,S,T and Y) which should provide enough data to do simple, quick comparisons. For example, are certain PTM modified residues enriched or depleted in certain mutation sub-types (disrupting / increasing etc) compared to their non-modified counterparts. More specifically, from supplementary Figure 3d we can see that when serine is mutated to aspartic acid, we see a higher normalized frequency of "increasing" interactions. Are these sites more often phosphorylated than we would expect by random chance?

Response to reviewers:

A new section has been added to the paper where we explore the overlap between phosphorylation sites and mutations in our data set. Other PTMs were also explored, but the overlap was too small to extract statistically sound conclusions. From our analysis it seems that phosphorylation sites are indeed enriched in our mutations data set. The results can be seen in the newly added figure 5.

2) With regards to contradictory results, the authors quote: “One reason for these contradictory results may be differences in experimental methodologies used to measure the effect, since IMEx databases recognize a large variety of experimental approaches that provide molecular interaction evidence.”

Do they not have the data to test this? At least on a fundamental principle of assay type. So do they come from immuno-precipitations versus two-hybrid. Or are the experimental differences more subtle?

Response to reviewers:

After careful examination of the contradictory cases, we found that most of them were actually experiments in which different proteoforms of the same reference protein were tested. For example, phosphorylated vs non-phosphorylated interacting partners tested for binding with mutant and “wild type” variants. These cases do not really represent a conflict in the effect of the variant and have been re-qualified in the current version of the manuscript as ‘consistent’. A revised version of figure 1f has been produced considering these cases as ‘consistent’. The criteria for defining conflicting mutations are described in detail in a new subsection in the ‘Methods’ section. We also detected a small number of conflicts arising from differences in curation criteria in legacy data. These have been manually corrected and re-annotated accordingly. The remaining ‘conflict’ cases are discussed in detail in the revised version of the manuscript. They are all caused by either different experimental approaches providing a different view of the data or by genuine conflicts between different publications.

Discussion points:

With more of an eye on the future, can the authors comment on how they are going to deal with deep scanning mutagenesis results, which they briefly allude to in the discussion. Specifically two points come to mind:

1) Will the database have standards for defining which mutants were tested in DMS approaches, as this could very strongly impact the “tested but no effect” category? In principle 10,000s of mutants can be synthesized en masse, but that doesn’t mean all are actually present in the libraries. Furthermore what is sequenced in the library input depends strongly on the sequencing error rate and number of reads. Will they leave it to each research group to define their mutant library coverage or set sequencing standards before the data can be submitted?

Response to reviewers:

IMEx curation policy states that we aim to capture the author's view of the publication and we refrain from making judgements on the data, which will already have been peer-reviewed. We have, however, set specific rules to deal with large-scale data sets in certain cases. Given that deep-mutagenesis approaches applied to the study of interactions are still relatively new, we have not yet defined a specific procedure for them as a Consortium. In this particular case, it seems reasonable to enforce a restrictive policy on the 'no effect' group of mutations, requiring that sufficient proof of presence of the mutant in the library needs to be provided for the annotation to be made.

2) The first comprehensive mutagenesis on both sides of a protein interaction has been undertaken to define both individual mutation impact and epistasis between mutations (Diss, Elife, 2018). It surely will not be the last. Can the authors say how / whether this database will deal with type of data (epistasis)?

Response to reviewers:

We thank the reviewer for bringing this study to our attention. The PSI-MI data model allows for links between different features (mutation annotations, in this case) and we will consider a new type of annotation to capture epistatic relations. We have added a comment in the discussion to this effect.

Minor comment:

X axis labels on figures 3d and 3e need shifting to the left.

Response to reviewers:

Affected figures have been corrected.

Reviewer #2 (Remarks to the Author):

The authors provided an open access resource created by IMEx database curators for describing the effects of mutations on protein interactions. The topic is very interesting, however, the manuscript just offers a descriptive analysis of the data set. It lacks deep biological knowledge and it is difficult to find how to use this dataset for functional investigation of the mutations in complex diseases. It may be more suitable for a more specialized database journal such as bioinformatics or Nucleic acids research.

Response to reviewers:

We suggest examples for applications of the data set in the improvement of variant effect prediction tools in our manuscript, which we hope illustrate one of the aspects in which its potential impact is more obvious. There is a solid base of literature studying the link between the impact of variation on the interactome and disease (Sahni et al, Cell, 2015 and Woodsmith et al, Nature Methods, 2017 are good examples) and more of this type of studies are to come. The IMEx mutations data set has the potential to become a key reference in the field, meriting publication in a journal with the broadest possible audience, such as Nature Communications.

1. The authors compared IntAct with UniProtKB and SKEMPI, but it is unknown how many of the curated records are consistent among different databases.

Response to reviewers:

In order to investigate record consistency, it was required to extend the comparison between the resources to be able to compare at the record level. Also, a new version of SKEMPI, SKEMPI 2.0, was published during the time this paper was in revision, so we updated our comparison table accordingly. In addition to providing a more detailed view of the differences between the resources, we have performed a detailed manual assessment of the curation consistency between the few publications that can be found in SKEMPI 2.0, UniProt mutagenesis annotations and the IMEx mutations data set. We found differences between the records that can be mostly attributed to different curation policies. A detailed commentary on these differences has been added to the revised manuscript.

2. It is not clear how many of these mutations are disease/cancer-related mutations, it would be better for the authors to provide the disease types.

Response to reviewers:

We have mapped diseases found to be associated with any IMEx-annotated mutations to the MeSH disease classification and re-done figure 3c using the MeSH classes instead of the diseases. We believe the new version of the figure is clearer and more informative. Supplementary table 2 has been added to provide the full list of mutation annotations-disease associations.

3. It is unknown whether these mutations are identified from low throughput experiments or computational predictions. The authors should compare whether the mutations from different sources will influence the conclusions.

Response to reviewers:

All the data in the data set comes from experimental evidence only, overwhelmingly from low throughput experiments (see figures 1c and 1d and general description of the data set at the beginning of the "Data set statistics" section). No predictions are included in the data.

4. The authors only used SIFT to evaluate the deleterious mutations, there are also many other tools and the accuracy of these tools are different. Other tools need to be compared to test the accuracy and the overall predictive value.

Response to reviewers:

We have performed additional analysis using the PolyPhen2 algorithm, a common method used for variant effect prediction. The results obtained confirm the deleterious mutations are significantly over-represented among those predicted to have extreme consequences according to this algorithm.

5. It is known that the majority of the protein interaction networks are biased to well-known proteins. However, it is unknown whether the curated data is biased for these proteins.

Response to reviewers:

The section entitled “Literature bias in the IMEx mutations data set” and figures 6a and 6b show that we have explored this bias and found that the data set is indeed slightly biased towards well-annotated proteins, as it would be expected from literature-curated annotations.

6. Are there any rules that can be learned from these curated mutations for developing computational methods to predict which mutations are more likely to perturb PPIs?

Response to reviewers:

We agree with the reviewer on the importance of using the data set to improve current variant effect predictors and have commented on its potential usage in the discussion section of the manuscript. However, extracting, using and validating such rules requires complex analysis and goes beyond the scope of this manuscript.

7. The mutations curated from different literature sources may be from different genome versions, how do the authors deal with this issue? Are the conclusions affected by this issue?

Response to reviewers:

As explained at the end of the section entitled ‘The IMEx mutations data set: data curation and quality control’ and in the Supplementary Methods, the data set is stored as part of the IntAct database, which is kept in sync with every UniProt release. Our monthly protein update pipeline ensures that sequences and annotations are updated with any changes in the UniProt reference. This process is overseen by expert curators and it ensures synchronization of our mutation annotations with current proteome builds. UniProt in turn keeps in sync with the latest genome builds as well, providing the mappings to genomic coordinates that we have used for some of the analysis in the manuscript. We plan to make this mapping part of the mutations data set routine release in the near future.

8. There are different types of mutations, including point mutations, insertion or deletion, but it is difficult to know which mutations were used in each section. The authors should classify the different mutation types clearly in the manuscript.

Response to reviewers:

Over 99% of the mutation cases we discuss are substitutions. At the end of the ‘Dataset statistics’ section we report that there are “only 65 deletion and 83 insertion annotations”, out of almost 28,000 annotations. Although we have analyzed the data set as a whole and deletions and insertions are included, we believe it reasonable to expect their impact on the results to be minimal and it should not merit -or even allow- for separate analysis.

9. The authors used a old version of the cBioPortal data. This version of the data is out of date since this database is updated by MSKCC quite frequently. The authors should check to see if any of the conclusions change significantly with these new data.

Response to reviewers:

Following the reviewer's recommendation, we have performed the analysis with the updated version of cBioPortal and the results are shown in figures 3f and 3g.

Reviewer #3 (Remarks to the Author):

In this paper the authors present the result of a titanic data curation effort to capture the impact of gene variation on protein-protein interactions. The final result is a database containing about 28,000 gene variations involving almost 14,000 protein interactions. The data is indeed very interesting, and it is worth being published, either in Nat Commun or somewhere else. Having said that, the manuscript is quite tedious, with long paragraphs describing the annotation process which, although very necessary, does not bring any new information. Additionally, from a more scientific perspective, the novelty of the manuscript is quite limited, since the authors apply a battery of very standard computational methods to the included variations (structure prediction, foldX, etc). At the end of the day, the value of the paper is the curated data, and I think it would be much clearer if the authors would just plainly describe the database, its contents and organization.

Response to reviewers:

We agree with the reviewer in that the core value of the paper is the curated data, so the analysis was limited to a descriptive study of the data using standards methodologies. The motivation behind it is to provide the users with an accurate picture of the data scope, strengths and limitations. We believe the analysis shown helps building this picture much more effectively than just a plain description of the data set contents and organization.

I am a bit puzzled by the results presented. The paper reports that, according to their annotations, only 11% of the aminoacid variations show no effect on the interaction patterns of the corresponding proteins. However, this figure is significantly higher in some of the main papers used to obtain and curate the data. For instance, Sadhi and co-workers (Cell 2015), which reported the effect of about 4,000 variations on the interaction profiles, saw that disease-causing mutations (i.e. more severe) cause little or no effect on 43% of the studied proteins. And this number goes up to almost 100% for non-disease mutations. Similar trends can also be derived from Yang and colleagues (Cell 2016), where they studied the interaction patterns of splice variants (i.e. protein isoforms). Obviously, they often involve much drastic changes than just a residue and, nevertheless, about 60% of the isoforms of the proteins studied only changed marginally (less than 50%) their interaction patterns.

I am thus surprised of the large impact on the interaction profiles reported in this paper. If they are indeed correct, the authors should discuss these apparent discrepancies.

Response to reviewers:

As discussed in the manuscript, the proportion of mutation annotations reporting no effect recorded in our data set is artificially low for two reasons: one, that authors often do not report results that show no effect versus a wild-type form; and two, that our own curation practices only recently have encouraged the capture of mutations that show no effect over interaction outcome. The proportion of such annotations should indeed be higher than those with an effect and we believe that will be soon the case in our data set, once studies systematically exploring all potential amino acid replacements along the sequence of a protein and their effect over interaction outcome are published.

REVIEWERS' COMMENTS:

Reviewer #1 (Remarks to the Author):

The authors have answered the comments and criticisms of the original manuscript, clarifying certain parts of the dataset and adding new analysis that add value to the interpreting the collated data. This should be a valuable asset to both computation and laboratory researchers alike.

Reviewer #2 (Remarks to the Author):

Major comment on page 14. Sahni et al; found only 2 gain of interaction AND not gain of function. Furthermore the sahani et al; paper found only 2 events of gain of interaction because they did not perform proteome wide screens for disease mutations due to the already large scale effort for screening of loss of interaction. Future efforts are underway to systematically screen for neomorphic mutations that cause gain of interaction and in turn gain of function.

Please update the text in the manuscript with the above recommendation.

Reviewer #3 (Remarks to the Author):

As I said in my previous assessment, the data presented is very valuable, and it definitely deserved being published. However, in my opinion, the authors have made no attempt to address my two comments, namely: i) too detailed information on the curation process, which has been described elsewhere and makes some parts of the m/s tedious and, more importantly ii) the numbers presented give the false idea that most (almost all) mutations will have an impact on protein interaction patterns, which is clearly wrong. In my opinion, saying that this might be an overestimation is not enough. All in all, since the m/s hasn't changed much, I stick to my original comments (which are, overall, positive !).

Reviewer's comments:

Reviewer #1 (Remarks to the Author):

The authors have answered the comments and criticisms of the original manuscript, clarifying certain parts of the dataset and adding new analysis that add value to the interpreting the collated data. This should be a valuable asset to both computation and laboratory researchers alike.

Response:

We thank the reviewer for their praise.

Reviewer #2 (Remarks to the Author):

Major comment on page 14. Sahni et al; found only 2 gain of interaction AND not gain of function. Furthermore the sahni et al; paper found only 2 events of gain of interaction because they did not perform proteome wide screens for disease mutations due to the already large scale effort for screening of loss of interaction. Future efforts are underway to systematically screen for neomorphic mutations that cause gain of interaction and in turn gain of function.

Please update the text in the manuscript with the above recommendation.

Response:

We thank the reviewers for this precision. The text has been updated with 'gain-of-function' replaced with 'gain-of-interaction'.

Reviewer #3 (Remarks to the Author):

As I said in my previous assessment, the data presented is very valuable, and it definitely deserved being published. However, in my opinion, the authors have made no attempt to address my two comments, namely: i) too detailed information on the curation process, which has been described elsewhere and makes some parts of the m/s tedious and, more importantly ii) the numbers presented give the false idea that most (almost all) mutations will have an impact on protein interaction patterns, which is clearly wrong. In my opinion, saying that this

might be an overestimation is not enough. All in all, since the m/s hasn't changed much, I stick to my original comments (which are, overall, positive !).

Response:

We believe that the curation process detail are required as reference for potential users to understand the representation procedure and the effects it has on the data set outline. Regarding the 'no effect' mutations, we clearly state in the text that they have only recently started to be annotated and that their numbers are not directly comparable with other effect categories.